# Compact Attention: Exploiting Structured Spatio-Temporal Sparsity for Fast Video Generation

## Abstract

The quadratic computational complexity of self-attention mechanisms pose a critical challenge for transformer-based video generation in synthesizing ultra-long sequences. Current sparse approaches with fixed patterns fail to fully exploit the inherent spatio-temporal redundancies in video data. Through systematic analysis of video diffusion transformers (DiT), we observed that critical information in attention matrices can be stably covered in a set of structured, yet heterogeneous, sparse patterns, including a cross-shaped attention pattern and patterns' dynamic behavior over time. To fully exploit the redundancies, we propose **Compact Attention**, a hardware-aware acceleration framework featuring three innovations: 1) Adaptive tiling strategies that approximate diverse spatial interaction patterns via dynamic tile grouping, 2) Temporally varying windows that adjust sparsity levels based on frame proximity, and 3) a recall-driven, offline search algorithm that automatically optimizes sparse masks while preserving critical attention pathways. Our method achieves up to 3x acceleration in Hunyuan and 2.25x in 14B Wan2.1 model in attention computation on single-GPU setups while maintaining comparable visual quality with full-attention baselines. By grounding acceleration in the empirical discovery of fundamental attention structures, this work provides a principled approach to efficient long video generation through structured sparsity exploitation.

## 1 Introduction

The rapid advancement of generative models has enabled high-quality video synthesis. However, processing ultra-long sequences remains a critical bottleneck. For Transformer-based video generation models, the quadratic complexity of self-attention mechanisms presents a fundamental challenge, as modeling spatiotemporal dependencies requires handling extensive token sequences. For example, in the Hunyuan-video architecture Kong et al. (2024), generating a 129-frame 720p HD video entails processing over 100K tokens, with attention computation consuming 68-72% of the total generation time. This computational burden becomes prohibitive for long-form video generation, necessitating the development of innovative acceleration strategies.

Recent studies Zhang et al. (2025c); Xia et al. (2025); Hassani et al. (2025); Xi et al. (2025); Ding et al. (2025); Zhang et al. (2025a); Yang et al. (2025c) show that full attention matrices in video generation exhibit significant sparsity, with complex distributions of attention weights and structured yet seemingly irregular patterns, indicating substantial untapped acceleration potential. The primary challenge lies in effectively leveraging these heterogeneous sparse patterns. However, even if accurate predictions can be made, the overhead associated with locating the sparse locations often offsets the potential speed gains.

During our analysis of video diffusion transformers (DiT), we observed a critical phenomenon: the interaction between 3D spatiotemporal token sequences may induce periodic, hierarchical attention patterns. As shown in Fig. 1, attention heads sometimes emerge with clear, distinct functional roles: focusing on local spatial regions, forming cross-shaped spatial interactions, and exhibiting a global or input-related focus. Additionally, certain heads exhibit relationships with frames at specific relevant distances. Some attention heads demonstrate temporal locality by suppressing distant frames,

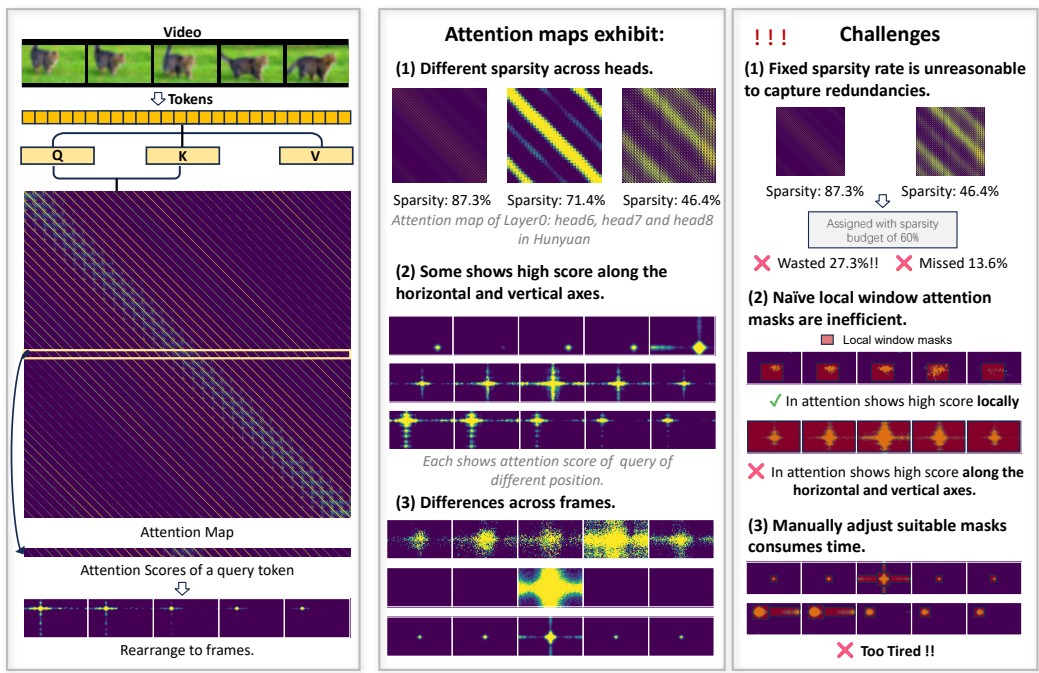

Figure 1: **The core motivation of Compact Attention.**

while others do the opposite. These structured sparsity patterns present opportunities for efficient approximation, however, existing methods fail to fully exploit these patterns.

Previous approaches, such as Minference Jiang et al. (2024), generalize sparse attention from language models by using fixed patterns (e.g., diagonals, blocks), but when applied to video generation, the method fail to exploit the unique 3D redundancies inherent in video data. Sparge Attention Zhang et al. (2025a) improves spatial grouping through Hilbert-order flattening and leverages locality, but it introduces additional overhead. SVG Xi et al. (2025) recognizes the unique periodic patterns in video data but does not account for the dynamic sparsity exhibited by each attention head. The sliding window approach in STA Zhang et al. (2025c) captures local spatiotemporal correlations but limits attention to rigid cubic regions, thereby missing crucial cross-frame interactions, sparsity related to relative temporal distance, and corresponding redundancies.

These limitations highlight the need for a video-specific sparse attention mechanism that can adaptively capture structured and dynamic spatiotemporal patterns. We summarize our key contributions as follows:

- We observed structured attention patterns in video diffusion transformers, whose specialized spatiotemporal attention behaviors motivate efficient sparse approximations.
- We propose **Compact Attention**, a training-free sparse attention framework that integrates an offline configuration search strategy with an efficient attention computation mechanism, while preserving the fidelity of generated videos.
- We validate our approach on the Wan2.1 and Hunyuan model, achieving up to **3×** attention speedup with negligible degradation in generated video quality.

## 2 RELATED WORKS

**Acceleration of diffusion models.** Due to the high inference cost, accelerating diffusion models has become a central research focus. Existing methods largely aim to reduce sampling steps and fall into two main categories: improved sampling algorithms Song et al. (2020); Lu et al. (2022); Liu et al. (2023); Bao et al. (2022); Liu et al. (2022); Zhang & Chen (2022) and distillation-based approaches Meng et al. (2023); Salimans & Ho (2022); Yin et al. (2024a); Li et al. (2024a); Xie

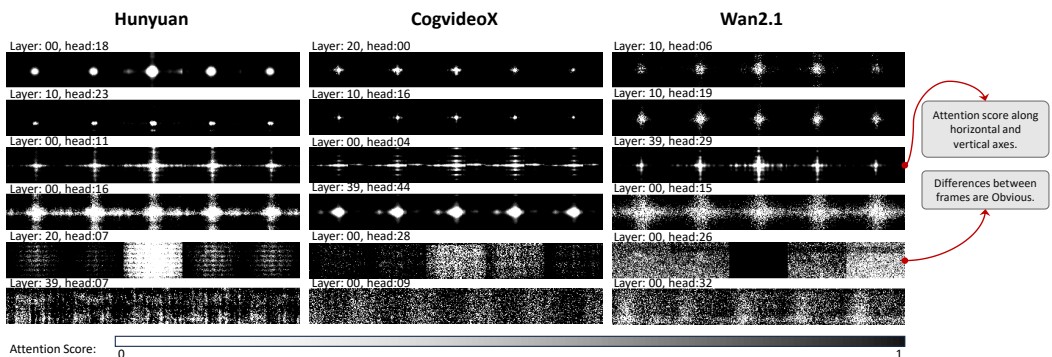

Figure 2: Visualization of characteristic attention patterns across different video transformers. Each map shows attention from a query token located in the center of the middle frame. The horizontal axis represents tokens flattened spatially within each frame, with frames themselves concatenated temporally. This layout reveals distinct spatiotemporal behaviors. The top row for each model shows a time-variant head attending across multiple frames, while the bottom row shows a time-invariant head focusing on a similar spatial pattern within its respective frame.

et al. (2024); Heek et al. (2024); Yin et al. (2024b); Sauer et al. (2024); Kim et al. (2023); Song et al. (2023). Distillation methods compress multi-step diffusion into a compact student model via teacher-student training, reducing inference steps. Beyond step reduction, several works Agarwal et al. (2024); Li et al. (2023); Lv et al. (2024); Kahatapitiya et al. (2024) explore cache mechanisms to eliminate redundant computation. Notably, Ma et al. (2024a) proposes a learning-based caching (L2C) strategy, while Wimbauer et al. (2024) applies block-level caching to reuse layer outputs across steps. DeepCache Ma et al. (2024b) further leverages temporal redundancy by reusing high-level features and updating only low-level ones. In addition to these approaches, attention-level optimization such as attention quantization and sparsification Zhao et al. (2024); Li et al. (2024b); Zhang et al. (2024b;a)—offers complementary acceleration potential.

**Sparse attention.** Sparse attention reduces the quadratic complexity of self-attention by masking computations to predefined sparse regions. In large language models, numerous studies (Zhu et al., 2024; Yang et al., 2025a; 2024; Xiao et al., 2023; Fu et al., 2025; Cai et al., 2025) have explored sparse attention designs. Some methods (Tang et al., 2024; Li et al.; Fu et al., 2024; Han et al., 2023; Xiao et al., 2024b; Zhang et al., 2023; Xiao et al., 2024a; Cai et al., 2024) use fixed patterns targeting specific positions and several works (Jiang et al., 2024; Ribar et al., 2023; Singhania et al., 2024; Lai et al., 2025; Gao et al., 2024) introduce input-adaptive sparse attention. In video generation, however, the inherent 3D redundancy poses challenges for directly transferring LLM-based sparse attention methods. Recent video-specific methods (Xia et al., 2025; Hassani et al., 2025; Xi et al., 2025; Ding et al., 2025; Zhang et al., 2025a; Yang et al., 2025c) introduce sparsity by masking the attention matrix based on observed patterns. These masks are either static and pre-defined (Zhang et al., 2025c; Li et al., 2025; Xi et al., 2025) or determined dynamically (Zhang et al., 2025b; Shen et al., 2025) during inference. Promising results have also been demonstrated by combining sparse attention with quantization (Liu et al., 2025; Zhao et al., 2025; Feng et al., 2025), distillation, and pre-training.

# 3  UNCOVERING STABLE ATTENTION PATTERNS FOR OFFLINE ATTENTION MASK PRECOMPUTATION

Our approach is founded on a foundational observation: attention maps show critical areas under a stable, structured set of patterns within video transformers. Uncovering and validating this finding allows for a principled, offline acceleration strategy. This section details several structured attention patterns we observed, and verifies their stability through large-scale statistical validation afterwards.

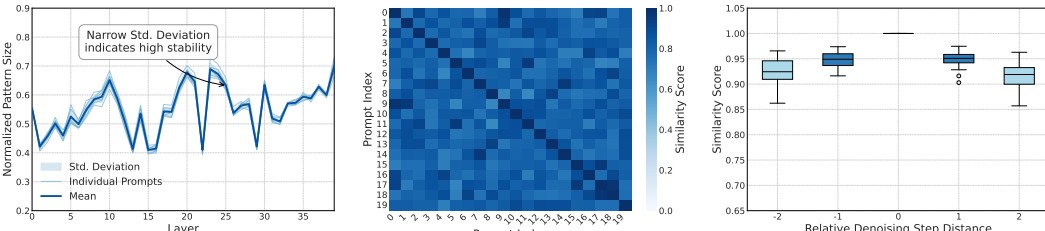

Figure 3: **Statistical validation of attention pattern stability.** **(a)** Mean similarity score of optimized masks across all layers, computed over 4,700 samples. The narrow standard deviation (shaded area) demonstrates high consistency regardless of input prompt or random seed. **(b)** Similarity heatmap between final masks generated from 20 distinct prompts. The uniformly high similarity scores confirm that pattern structures are largely input-agnostic. **(c)** Box plot of mask similarity across different relative denoising step distances. The consistently high median similarities (all 0.9) validate the temporal robustness of the patterns, enabling mask reuse.

## 3.1 STRUCTURED SPATIOTEMPORAL PATTERNS IN ATTENTION MAPS

The attention maps derived from 1D sequences with 3D structural information (f, h, w) exhibit highly complex diversity. Existing sparse attention methods in video generation models have identified clustering patterns such as slash-line, vertical-line, and block-shaped formations in these attention maps. However, a fine-grained analysis of per-query-token attention distributions facilitates a structured interpretation of the intricate patterns within the attention maps.

By examining full attention maps at the query-specific token level, we observe that diagonal patterns arise from systematic position-relative attention biases, which visually manifest as distinct spatiotemporal modes. Our empirical analysis reveals a recurring set of such patterns. Among the most prominent and potentially efficient are three spatial patterns and two distinct temporal behaviors that are commonly present in 3D full-attention video generation models (as illustrated in Fig. 2).

**Spatial Patterns:**

- *Local Patterns*: Certain attention heads focus on compact neighborhoods around target positions, forming spherical attention fields that are likely crucial for fine-grained detail synthesis.
- *Cross-Shaped Patterns*: Specialized attention heads exhibit directional sensitivity, creating continuous attention corridors along the horizontal and vertical axes.
- *Global Patterns*: Some attention heads preserve full spatial connectivity irrespective of the relevant distance. Additionally, input-dependent attention heads exhibit strong weight clustering around salient objects, which are also observed as global patterns.

**Temporal Patterns:**

- *Time-Variant Patterns*: This pattern exhibits a strong correlation with temporal relative distance. Some attention heads demonstrate progressive weight decay across frames, while others focus more on frames at a specific distance, excluding local or nearby frames.
- *Time-Invariant Patterns*: These attention heads maintain frame-agnostic distributions, ensuring a consistent focus across all timesteps regardless of the relative temporal distance.

## 3.2 PATTERN STABILITY ENABLES OFFLINE ACCELERATION

A key finding that enables our entire approach is that the patterns we identified are not ephemeral but exhibits remarkable stability, making an offline pre-computation strategy feasible. We confirmed this through a large-scale statistical analysis across over 4,700 samples (derived from 946 diverse prompts, each with 5 random seeds) on Wan2.1 and Hunyuan.

**Input/Seed Invariance.** Our systematic analysis demonstrates that these spatiotemporal attention patterns arise as inherent properties of the model architecture, rather than being driven by input adaptations. As demonstrated in Fig. 3b, the final derived sparse masks stay alike with an average

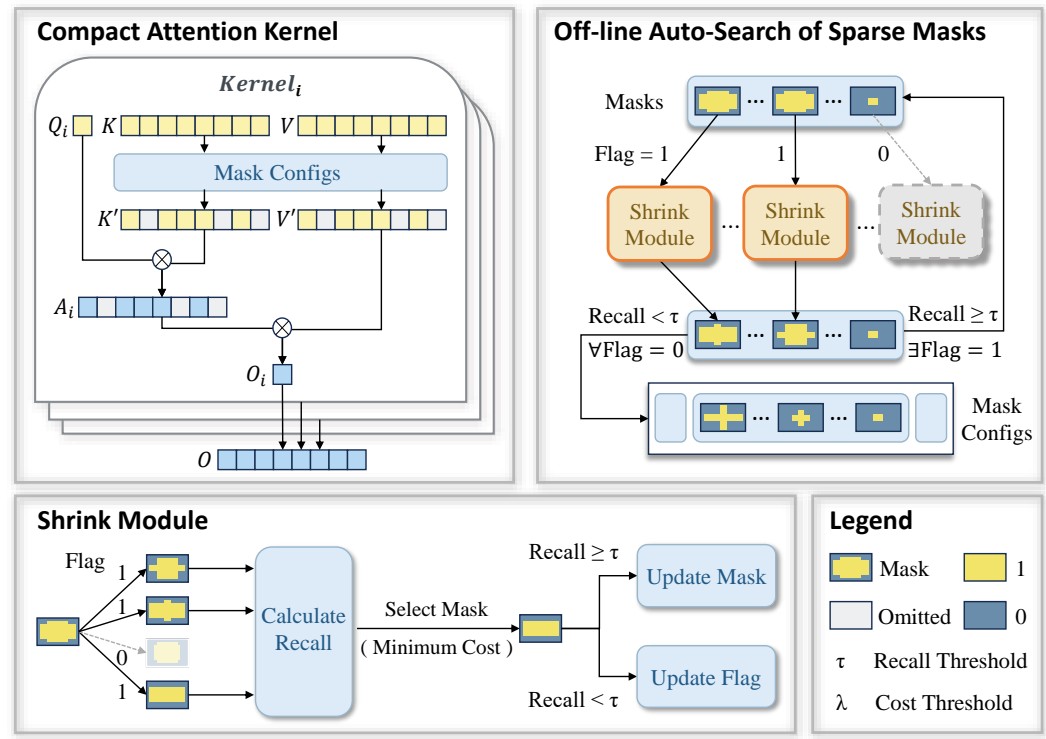

Figure 4: Compact Attention: Pipeline. (Right) Auto-Search: Iteratively shrinks attention windows based on cost threshold and recall threshold. Flags indicate mask status (1=active, 0=frozen). (Left) Kernel: Applies the optimized sparse configurations to accelerate generation.

similarity score over 0.8 across varying text prompts and random initializations, measured by similarity between masks gained from merging and masks searched under different prompts or seeds:

$$\mathrm{Sim}(M_A, M_B) = \frac{\|M_A \odot M_B\|_1}{\|M_A + M_B - M_A \odot M_B\|_1} \tag{1}$$

where $M_A, M_B$ are binarized attention masks. The average similarity was high across different prompts (81.24% for Wan2.1, 84.25% for Hunyuan), as was the similarity across different seeds (82.74% for Wan2.1, 85.23% for Hunyuan). This high consistency is the foundation of our offline search.

**Temporal Robustness.** The patterns are also stable across the denoising process. As shown in Fig. 3c, attention configurations within a certain range of denoising steps are highly similar, enabling reliable reuse of pre-computed masks.

## 4 COMPACT ATTENTION

Based on the observation of the special attention patterns in Section 3, we introduce Compact Attention, a training-free framework designed to exploit these structures for hardware-efficient acceleration. The framework consists of two main components: a novel Deformable Sparse Pattern mechanism that can flexibly represent the discovered patterns, and a recall-driven Auto-Search Algorithm that optimizes these patterns offline. The entire pipeline is illustrated in Fig. 4.

### 4.1 TILE-BASED DEFORMABLE SPARSE PATTERN

To effectively capture the complex attention structures—including local, cross-shaped, and temporal variations—we propose a deformable sparse pattern mechanism that adapts across both spatial and temporal dimensions. Our approach is fundamentally tile-based to ensure hardware efficiency. Following the insight from Zhang et al. (2025c), we first reorganize video tokens into spatially contiguous tiles. Details can be found in Appendix. E. Our sparse masks then operate at the granularity

Figure 5: Compact Attention: Pipeline

of these tiles. Building on this foundation, Compact Attention introduces two key innovations: 1) **Dual Attention Windows:** To model the diverse spatial patterns (Local vs. Cross-Shaped) without inefficient pre-classification, we define the search space for each spatial mask as the union of two independent rectangular windows. This simple yet powerful design allows the offline search algorithm (described next) to flexibly form various shapes. It can create a local pattern by converging the two windows into one, or naturally form a cross-shaped pattern by optimizing one window to be tall and thin, and the other to be short and wide. This adaptability is the key to efficiently capturing each head's unique structure. 2) **Frame-Group-wise Configurations:** To model the observed temporal dynamics (Time-Variant vs. Time-Invariant patterns), we partition frames into distance-based groups relative to the current query frame. Each group is assigned its own independent sparse configuration, allowing for aggressive sparsity on less-correlated frames while preserving detail for strong-correlated ones.

This deformable architecture achieves a three-fold synergy: (1) spatial adaptability through tile combinations that emulate diverse attention modes, (2) temporal awareness via distance-stratified configurations, and (3) hardware efficiency by preserving the computational regularity inherent in tile-based processing.

### 4.2 Off-line Auto-Search of Sparse Masks

Leveraging the pattern stability validated in Section 3, we could prepare sparse masks for each model in advance via an offline auto-search algorithm. This addresses the prohibitive overhead of online prediction. The algorithm is designed as a cost-affordable, greedy search process, as detailed below and in Appendix. F. The whole pipeline is shown in Fig. 4.

**Mask Contraction Process:** Guided by the spatial and temporal variation characteristics presented in Section 3, We formulate mask optimization as a boundary contraction process along hierarchical dimensions. The process starts with full attention coverage and iteratively tightens window boundaries across spatial dimensions, prioritizing regions with lower recall contributions, as indicated by the recall loss per computational unit (cost). As shown in 5, This directional shrinkage operates independently across different frame groups. As shown in Fig. 4, A binary flag is used to track the status of each mask group; only active masks (Flag=1) are considered for further shrinking.

**Dual-Threshold Governance:** The contraction process is governed by dual thresholds: a minimum recall threshold $\tau$, which preserves critical interactions, and a maximum cost threshold $\lambda$, which balances computational reduction against accuracy loss. The mask shrinking process for a given frame group terminates when either the recall drops below $\tau$ or the cost of further shrinking ($\Delta Recal/\Delta Cost$) exceeds $\lambda$, ensuring an effective trade-off between quality and efficiency. To obtain the final configuration, we merge configurations across prompts through union operation. This conservative merging strategy guarantees that all potentially relevant attention regions are retained. The core algorithm logic of the Cost-Effective Greedy Window Search algorithm is shown in Appendix. F. **Reuse Masks across Denoising Steps:** Capitalizing on the similarity of sparse masks searched within a certain denoising step range, we implement mask reuse across $n$ consecutive denoising steps, which reduces the search frequency by $n\times$, while maintaining generated video quality. In our implementation, we update the sparse mask configuration every 5 denoising steps. Within each interval, the mask derived from the first step is reused.

Table 1: Quantitative Comparison of Sparse Attention Methods in Wan2.1 and Hunyuan on VBench: Visual Consistency, Aesthetic Quality and Text-Video Alignment.

| Model (seq_len) | Method | Sparsity | Subject Consistency | Background Consistency | Aesthetic Quality | CLIPSIM | CLIP-T |
|---|---|---|---|---|---|---|---|
| Wan2.1 Wan et al. (2025) (80K) | FlashAttention-2 Dao et al. (2022) | 00.00% | 0.9709 | 0.9721 | 0.6777 | 0.1933 | 0.9989 |
| | FlashAttention-3 Shah et al. (2024) | 00.00% | 0.9681 | 0.9616 | 0.6486 | 0.1944 | 0.9985 |
| | Sparse VideoGen1 Xi et al. (2025) | 25.02% | 0.9497 | 0.9611 | 0.6273 | 0.1939 | 0.9985 |
| | Sparse VideoGen2 Yang et al. (2025b) | 28.10% | 0.9524 | 0.9612 | 0.6351 | **0.1949** | 0.9985 |
| | SpargeAttn Zhang et al. (2025a) | 32.27% | 0.9357 | 0.9500 | 0.5320 | 0.1891 | 0.9982 |
| | **Compact Attention (Ours)** | 33.99% | 0.9659 | **0.9650** | **0.6480** | 0.1901 | 0.9985 |
| | **Compact Attention (Ours)** | 24.66% | **0.9674** | 0.9638 | 0.6459 | 0.1929 | **0.9986** |
| Hunyuan Kong et al. (2024) (127K) | FlashAttention-2 Dao et al. (2022) | 00.00% | 0.9741 | 0.9736 | 0.6665 | 0.1984 | 0.9995 |
| | FlashAttention-3 Shah et al. (2024) | 00.00% | 0.9736 | 0.9735 | 0.6542 | 0.2015 | 0.9995 |
| | Sparse VideoGen1 Xi et al. (2025) | 62.20% | 0.9495 | 0.9660 | 0.6185 | 0.2038 | 0.9993 |
| | Sparse VideoGen2 Yang et al. (2025b) | 58.88% | 0.9553 | 0.9663 | 0.6232 | **0.2134** | 0.9992 |
| | SpargeAttn Zhang et al. (2025a) | 47.77% | 0.9664 | 0.9731 | 0.5794 | 0.1961 | 0.9995 |
| | **Compact Attention (Ours)** | 62.36% | 0.9716 | 0.9693 | 0.6531 | 0.2010 | 0.9995 |
| | **Compact Attention (Ours)** | 52.90% | **0.9723** | **0.9735** | **0.6536** | 0.2014 | 0.9995 |

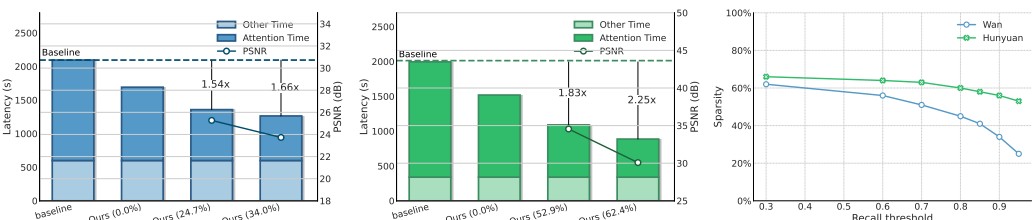

Figure 6: (a) End-to-end acceleration of Wan2.1. (b) End-to-end acceleration of Hunyuan. (c) Sparsity trends under different recall thresholds for auto-searched attention patterns on Wan2.1 and Hunyuan.

## 5 EXPERIMENTS

### 5.1 EXPERIMENTAL SETUP

Our evaluations are primarily conducted on the state-of-the-art video generation architecture Wan2.1(14B) Wan et al. (2025) and Hunyuan Kong et al. (2024) on a single H800 GPU. We apply Compact Attention to generate outputs consisting of 81 frames in Wan2.1 (seq_len = 80k) and 129 frames in Hunyuan (seq_len = 127k) at a resolution of 768×1280. To evaluate the acceleration effect achieved through the exploitation of attention sparsity by Compact Attention, we measured video quality using SSIM, PSNR, MSE and six quality metrics (Subject Consistency, Background Consistency, Aesthetic Quality) in VBench, and CLIPSIM and CLIPTemp (CLIP-T) Liu et al. (2024) to measure the text-video alignment on Open-Sora benchmark. For computational performance, we report both the attention sparsity rate, end-to-end latency and attention latency. Compact Attention is implemented based on ThunderKittens and FlashAttention3, with reference to the STA framework.

**Baselines:** We evaluated Compact Attention against several state-of-the-art sparse attention approaches, including STA Zhang et al. (2025c)(spatio-temporal locality), Sparse VideoGen Xi et al. (2025)(static pattern), Sparse VideoGen2 Yang et al. (2025b)(dynamic pattern) and Sparse Attention Zhang et al. (2025a)(dynamic pattern). For performance comparison, we measured similarity relative to Full attention to show quality preserved by sparse attention mechanism, and quantified speedup additionally with FlashAttention-2(Dao et al. (2022)). Additional implementation details can be found in Appendix. C.

### 5.2 ACCELERATION PERFORMANCE AND QUALITY PRESERVATION

**Quality Evaluation** In some cases, video similarity with full attention is not meaningful to assess, as outputs from certain sparse attention variants deviate from the original content, however maintain

Table 2: Comparative Analysis of Sparse Attention Efficiency. Compact Attention achieves high-quality video generation leveraging better mask design.

| Model (seq_len) | Method | Sparsity | Quality | | | Speed | |
|---|---|---|---|---|---|---|---|
| | | | SSIM ↑ | PSNR ↑ | MSE ↓ | Latency (s) | Speedup |
| Wan2.1 Wan et al. (2025) (80K) | FlashAttention-2 Dao et al. (2022) | 00.00% | - | - | - | 1496.7s | 1.00x |
| | FlashAttention-3 Shah et al. (2024) | 00.00% | - | - | - | 795.63s | 1.88x |
| | Sparse VideoGen1 Xi et al. (2025) | 25.02% | 0.6642 | 18.7342 | 1128.257 | 1107.9s | 1.35x |
| | Sparse VideoGen2 Yang et al. (2025b) | 28.10% | 0.7019 | 19.4262 | 762.2486 | 1118.8s | 1.34x |
| | SpargeAttn Zhang et al. (2025a) | 32.27% | 0.6628 | 19.6564 | 806.9727 | 1065.8s | 1.40x |
| | **Compact Attention(Ours)** | 33.99% | _0.7382_ | _20.9106_ | _715.9552_ | **663.82s** | **2.25x** |
| | **Compact Attention(Ours)** | 24.66% | **0.7650** | **21.7350** | **637.7155** | _758.18s_ | _1.97x_ |
| Wan2.1 Wan et al. (2025) (69K) | FlashAttention-2 Dao et al. (2022) | 00.00% | - | - | - | 1106.0s | 1.00x |
| | FlashAttention-3 Shah et al. (2024) | 00.00% | - | - | - | 580.44s | 1.91x |
| | STA Zhang et al. (2025c) | 63.48% | 0.7545 | 20.6110 | 777.4463 | **333.66s** | **3.31x** |
| | **Compact Attention(Ours)** | 33.99% | _0.7754_ | _23.7297_ | _351.6015_ | _481.86s_ | _2.30x_ |
| | **Compact Attention(Ours)** | 24.66% | **0.8147** | **25.2664** | **254.1789** | 539.86s | 2.05x |
| Hunyuan Kong et al. (2024) (127K) | FlashAttention-2 Dao et al. (2022) | 00.00% | - | - | - | 1655.2s | 1.00x |
| | FlashAttention-3 Shah et al. (2024) | 00.00% | - | - | - | 951.32s | 1.74x |
| | Sparse VideoGen1 Xi et al. (2025) | 62.20% | 0.7472 | 21.3601 | 619.5743 | _633.62s_ | _2.61x_ |
| | Sparse VideoGen2 Yang et al. (2025b) | 58.88% | 0.8579 | 26.8913 | 211.2457 | 724.58s | 2.28x |
| | SpargeAttn Zhang et al. (2025a) | 47.77% | 0.7787 | 23.5714 | 370.4673 | 1148.6s | 1.44x |
| | **Compact Attention(Ours)** | 62.36% | _0.8636_ | _27.4991_ | _165.4867_ | **546.50s** | **3.03x** |
| | **Compact Attention(Ours)** | 52.90% | **0.9159** | **31.2729** | **82.38691** | 750.20s | 2.21x |
| Hunyuan Kong et al. (2024) (115K) | FlashAttention-2 Dao et al. (2022) | 00.00% | - | - | - | 1304.0s | 1.00x |
| | FlashAttention-3 Shah et al. (2024) | 00.00% | - | - | - | 723.93s | 1.80x |
| | STA Zhang et al. (2025c) | 58.37% | 0.8385 | 26.4213 | 182.3809 | **413.96s** | **3.15x** |
| | **Compact Attention(Ours)** | 62.36% | _0.9040_ | _30.0822_ | _105.1957_ | _476.45s_ | _2.74x_ |
| | **Compact Attention(Ours)** | 52.90% | **0.9452** | **34.5506** | **35.13070** | 548.18s | 2.38x |

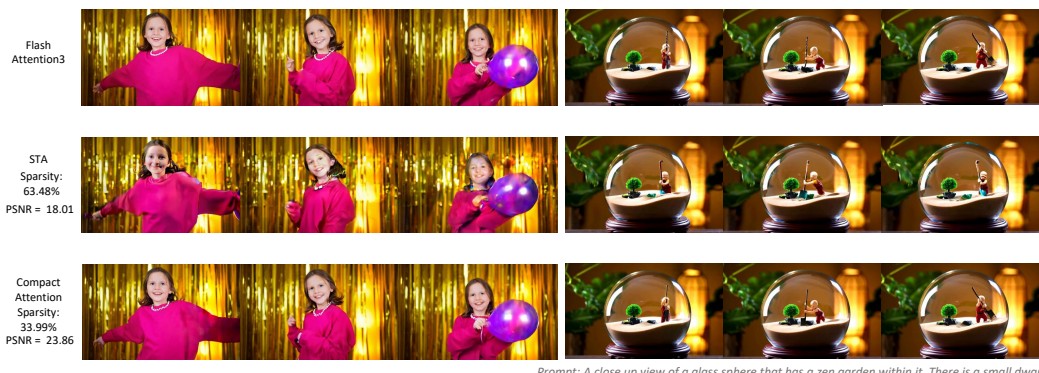

Prompt: A cartoon kangaroo disco dances.

Prompt: A close up view of a glass sphere that has a zen garden within it. There is a small dwarf in the sphere who is raking the zen garden and creating patterns in the sand.

Figure 7: Performance of different sparse attention methods on end-to-end video generation.

visual quality. Table 1 evaluates performance of sparse attention methods on VBench metrics. Our proposed method also outperform other sparse attention method at higher sparsity at a significant speedup ratio as reported also in 6. Visual comparisons in Figure 7 corroborate these findings.

**Similarity Evaluation** For not generating videos under recommended resolution may damage the quality of video and STA works on constrained resolution, we additionally compare video similarity with full attention under settings of 69 frames in Wan2.1 and 117 frames in Hunyuan. Tab. 2 illustrates the sparsity efficiency of Compact Attention on critical information preservaing within the Wan2.1 and Hunyuan model. Although achieving significant speedup, STA suffered from notable quality degradation through sparse mechanism. Analysis with STA shows that Compact Attention maintains high-quality generation in Hunyuan at a higher sparsity level (62.36%), as reflected by a average PSNR of **30.0822**.

## 5.3 ABLATION STUDIES

**Effectiveness of deformable mask design** To validate the effectiveness of our proposed tile-based mask design in exploiting more sparsity for 3D full attention, we conduct ablation experiments on different type of attention heads at a fixed params $\tau = 0.9$ and $\lambda = 0.011$ in mask searching on Wan2.1. Results are shown in Table 3. The experiment starts with a basic Cubic Window (sliding window with various sizes). We then progressively add our temporal-aware configuration (+ Frame-group-wise) and finally our adaptive spatial windows (+ Dual Windows), which constitutes the full Compact Attention design. The results clearly demonstrate that introducing frame-group-wise configurations increases sparsity on time-variant heads by 3%. The final addition of Dual Attention Windows provides a further, substantial 9.8% increase in overall sparsity.

Table 3: Ablation study on the sparse mask design. Each component progressively increases the achievable sparsity.

| Method | Locality | Cross | Global | Time-Variant | Time-Invariant | Overall Sparsity |
|--------|----------|-------|--------|--------------|----------------|------------------|
| Cubic Window (Baseline) | 0.726 | 0.385 | 0.078 | 0.441 | 0.306 | 0.361 |
| + Frame-group-wise | 0.758 | 0.406 | 0.085 | 0.472 | 0.317 | 0.370 **(+0.9%)** |
| + Dual Windows (Ours) | 0.766 | 0.516 | 0.099 | 0.567 | 0.385 | 0.459 **(+9.8%)** |

Table 4: Comparison of mask searching methods. Our method finds configurations with both higher sparsity and better quality.

| Model | Search Strategy | Sparsity (%) ↑ | PSNR (dB) ↑ |
|-------|-----------------|----------------|-------------|
| Wan2.1 | Pre-selected Set | 20.55 | 18.52 |
|  | **Greedy Search (Ours)** | **33.99** | **23.73** |
| Hunyuan | Pre-selected Set | 54.71 | 21.00 |
|  | **Greedy Search (Ours)** | **62.36** | **30.08** |

**Greedy Search Strategy.** To demonstrate the necessity of greedy search, we compare it against method used in STA: searching from a small, pre-selected set of fixed mask configurations. The pre-selected set includes the top 15 frequent sparse masks appeared in greedy mask searching. As shown in Table 4, while the pre-selected set offers some sparsity, greedy search algorithm consistently discovers configurations that are both sparser and yield higher-quality outputs (as measured by PSNR), as a greedy search can finely tailor the mask shape to each head's unique attention distribution, a level of optimization that a limited set of fixed templates cannot achieve.

**Sensitivity to Recall Threshold in Auto-Search.** Our auto-search is governed by the recall threshold ($\tau$) and cost threshold ($\lambda$), which controls the trade-off between acceleration and fidelity. Figure 6 illustrates this relationship on both Wan2.1 and Hunyuan. As expected, a higher recall threshold (e.g., 0.95) leads to lower sparsity as more attention connections are preserved to ensure higher fidelity. Conversely, relaxing the threshold to 0.9 allows for significantly greater sparsity and acceleration.

**Sensitivity to Sparsification in Early Denoising Steps.** We also observed that the content of generated videos are most sensitive to sparse attention during the early stages of the whole denoising process, where high-noise are. Quantitative results show a 1.02dB PSNR drop when full attention is applied only in the final 15 steps, compared to the first 15. This highlights the importance of preserving full attention in the early timesteps to ensure quality, while allowing sparsification in later stages for acceleration without compromising visual fidelity. As is partly shown in Fig. 8, our empirical results suggest that maintaining full attention for the initial 15 denoising timesteps proves essential for preserving generation quality, whereas applying sparse attention in the remaining steps achieves notable acceleration with minimal quality loss.

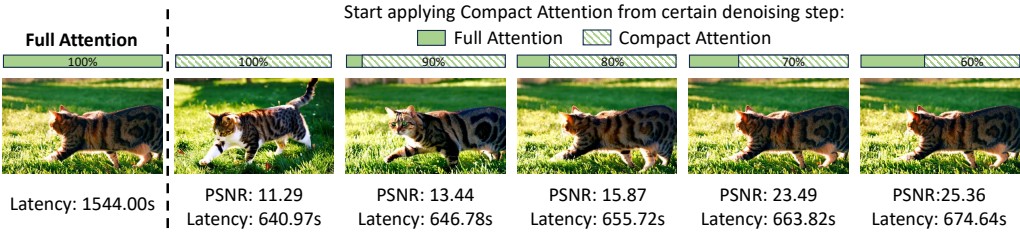

Figure 8: Effect of delaying sparse attention application: PSNR score and visual performance of sparse attention versus full attention.

## 6 CONCLUSION

The high computational cost of video generation models necessitates efficient attention mechanism that maintains generation quality. Based on the observation of the spatiotemporal attention patterns, including the cross-shaped and time-variant structures widely shown within video DiT models, We propose Compact Attention. This framework leverages flexible tile-based sparse masks and an automated mask search algorithm to accelerate attention computation by up to 2.5× while preserving visual quality. This work shares an interesting observation of special attention patterns and offers a practical, training-free approach to efficiently exploit them, unlocking a principled pathway for efficient long-form video generation, particularly in resource-constrained environments.

## REPRODUCIBILITY STATEMENT

All findings presented in this paper are fully reproducible. Pseudocode for our algorithm is available in the Appendix. Core codes are also provided in the supplementary materials. A detailed description of the experimental setup can be found in Section 5.1 and the methodology is comprehensively explained and supplemented with a figure to illustrate the workflow. We are confident that, with the provided resources, readers will be able to reproduce all of the results presented.

## LLM USAGE

Large Language Models(LLMs) were not used to aid in the writing or research in the paper.

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

## A   LIMITATIONS

The proposed auto-search strategy offers computational efficiency. However, through low recall and cost thresholds, this design choice may potentially compromise the visual fidelity of generated video content. Specifically, under more demanding scenarios, critical visual details could be omitted, leading to suboptimal generation quality. Future work will explore adaptive thresholding and context-aware search strategies to better balance efficiency and perceptual performance.

## B   BROADER IMPACTS

Our work reduces computational barriers for deploying long-video generation models through accelerated inference and lower memory costs, enabling broader access to high-quality video synthesis for individual creators and small teams. This democratization could catalyze innovation in education, digital art, and low-resource creative industries. Notably, our discovery of hierarchical attention patterns—such as localized spatial focus(local pattern, cross-shaped pattern), temporally-varying frame dependencies provides new insights into how video Transformers model spatiotemporal relationships. These patterns reveal specialized roles of attention heads (e.g., handling short-term motion or global context), improving model interpretability and offering a foundation for future research. Such findings could inspire targeted architectural designs (e.g., hybrid sparse attention modules) or curriculum learning strategies that align training with inherent spatiotemporal priors, potentially advancing both efficiency and controllability in video generation systems.

## C   BASELINE IMPLEMENTATION DETAILS

**STA**   STA is implemented based on FlashAttention-3 within the ThunderKittens framework and is compatible exclusively with the Hopper architecture. We adopt the publicly released mask configuration for Wan2.1 and Hunyuan from the official STA repository. Due to STA's strict constraints on video resolution, all experiments are conducted on 69-frame or 117-frame videos at a resolution of 768×1280 using the STA kernel for fair comparison.

**Sparge Attention**   Sparge Attention provides an interface for sparse attention operations via its open-source implementation. In our experiments, we integrate this interface into the `diffusers` library and evaluate the method using its default hyperparameters (`simthreshd1=0.1`, `cdfthreshd=0.9`, `pvthreshd=20`). The observed average sparsity is comparable to that of other baseline methods.

**Sparse VideoGen and Sparse VideoGen2**   We conduct experiments using the official implementation of Sparse VideoGen (SVG) from its open-source repository. The observed average sparsity is adjusted to be comparable to that of other baseline methods.

## D   COUNTER-INTUITIVE IMPROVEMENT

An interesting phenomenon shows that both Sparse VideoGen and our Compact Attention occasionally outperform the full-attention baseline on certain perceptual metrics. While videos from SpargeAttn exhibit low visual quality, both Sparse VideoGen and Compact Attention even outperform the full attention baseline on some metrics. This counter-intuitive improvement is could likely be interpreted as a "noise filter". By pruning non-essential attention connections, our method may filter unnecessary noises, discouraging the model from focusing on spurious correlations and forcing it to rely on more robust, structural dependencies, thereby shows this counter-intuitive improvement.

## E   SPARSITY VALIDATION AFTER REARRANGED BASED ON ADJACENT 3D TILES

FlashAttention tiles the query, key, and value tensors along the token dimension into blocks $Q_i$, $K_i$, $V_i$ with block sizes $b_q$, $b_k$ respectively, and computes each output block $O_i$ incrementally using

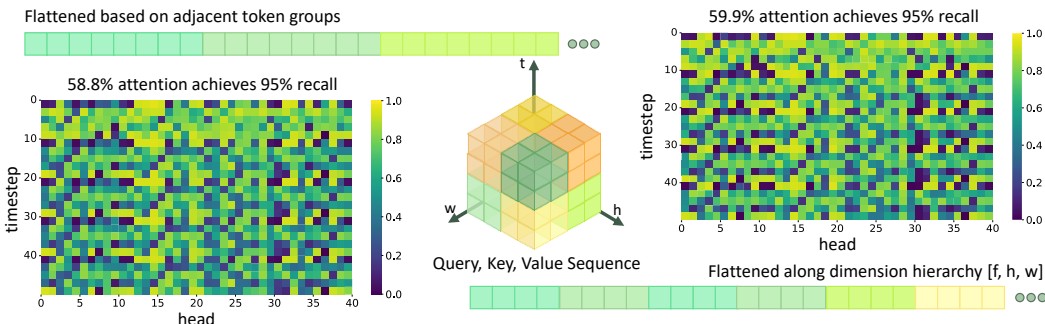

Figure 9: Heatmap of attention map and the $k\%$ values required to retain top-$k$ for 0.95 recall before and after rearranging attention maps into 3D spatially adjacent groups.

an online softmaxDao et al. (2022). This design achieves lower memory consumption and faster execution, while enabling attention acceleration through tile-level sparsity, thus avoiding the inefficiency of token-level sparsity. As a result, many sparse attention methods adopt small blocks as the basic computation unit. However, applying block-wise sparsity directly on attention over sequences obtained by flattening a 3D feature map (f, h, w) may be suboptimal, as it treats tokens within each block as equally important, regardless of spatial relationships. In Fig. 10 and Fig. 11, We show that reordering tokens based on 3D spatial locality prior to applying block-wise sparsity also improves attention sparsity while maintaining acceleration benefits in hunyuan. This spatially-aware grouping yields a **1%** reduction in the average number of active blocks on the Wan2.1 (14B) model, and **3.4%** on the Hunyuan model.

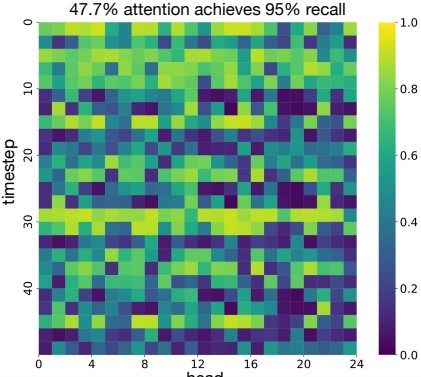

Figure 10: Flattening sequence on tiles    Figure 11: Directly flattening sequence

## F    ALGORITHM DETAILS

Here we provide our presudo code of the greedy search algorithm as follows 0.

## G    COST AND CONVERGENCE ANALYSIS OF OFFLINE SEARCH

Regarding the computational overhead of our offline search strategy, we provide a detailed breakdown of the time cost and analyze the convergence behavior of mask quality with respect to the number of proxy prompts.

**Computational Cost.** The offline search is a one-time pre-computation cost per model resolution. On a node with 8 NVIDIA H800 GPUs: For Wan2.1-14B (720p), the search process takes approximately 1.5 hours using parallel processing. For HunyuanVideo (720p), the search takes approximately 2 hours.

---

**Algorithm 1** Cost-Effective Greedy Window Search

---

1: **Input:**
2:    $Q, K, V$: Query, Key, Value tensors.
3:    $R_{\text{target}}$: The target attention recall threshold (e.g., 0.9).
4:    $E_{\text{thresh}}$: The cost-effectiveness threshold (max cost per unit of recall gain).
5:    $S$: The set of all possible single-step window shrinks.
6: **Output:**
7:    $C_{\text{best}}$: The final, optimized sparse attention configuration.

8: **Helper** Evaluate($C$):
9:    Computes attention recall $R$ and computational Cost for a given configuration $C$.
10:    **Return** $(R, \text{Cost})$.

11: **procedure**
12:    $C_{\text{best}} \leftarrow C_{\text{initial}}$                                           ▷ Start with the largest window
13:    $(R_{\text{last}}, \text{Cost}_{\text{last}}) \leftarrow \text{Evaluate}(C_{\text{best}})$
14:    **while** $R_{\text{last}} < R_{\text{target}}$ **do**
15:        PotentialMoves $\leftarrow \emptyset$        ▷ Find the most cost-effective shrinks across all dimensions
16:        **for** each shrink $e \in S$ **do**
17:            $C_{\text{candidate}} \leftarrow \text{ApplyShrinking}(C_{\text{best}}, e)$
18:            $(R_{\text{cand}}, \text{Cost}_{\text{cand}}) \leftarrow \text{Evaluate}(C_{\text{candidate}})$
19:            $\Delta R \leftarrow R_{\text{cand}} - R_{\text{last}}$
20:            $\Delta \text{Cost} \leftarrow \text{Cost}_{\text{cand}} - \text{Cost}_{\text{last}}$
21:            **if** $\Delta \text{Cost} > 0$ **then**
22:                $E \leftarrow \Delta R / \Delta \text{Cost}$                              ▷ Calculate Cost-Effectiveness
23:                add $(E, C_{\text{candidate}})$ to PotentialMoves
24:        **if** PotentialMoves $= \emptyset$ **then**
25:            **break**
                                                            ▷ Select the best move if it meets the threshold
26:        $(E_{\text{best}}, C_{\text{next}}) \leftarrow \text{argmin}_E(\text{PotentialMoves})$        ▷ Find move with best (lowest)
    cost-effectiveness
27:        **if** $E_{\text{best}} \leq E_{\text{thresh}}$ **then**
28:            $C_{\text{best}} \leftarrow C_{\text{next}}$
29:            $(R_{\text{last}}, \text{Cost}_{\text{last}}) \leftarrow \text{Evaluate}(C_{\text{best}})$
30:        **else**
31:            **break**                                           ▷ No more cost-effective moves available
32:    **return** $C_{\text{best}}$

---

Considering that training these models requires thousands of GPU-hours and deployment serves millions of requests, this one-time overhead is negligible (amortized to near-zero). The search only needs to be re-executed if there is a significant change in model architecture, aspect ratio (e.g., changing from landscape to vertical), or after major fine-tuning that alters the model's structural bias.

**Proxy Prompts Selection.** To effectively get robust static masks, we choose prompts whose video contents does not contain specific object to alleviate bias in searching. We chose prompts like "A red background.". And surprisingly, we found that models while sometimes do great in complex motion generation, fail to obey simple prompts like a background with one color.

**Convergence of Mask Union.** We employ a "Union" strategy to merge masks searched from N different proxy prompts. As shown in Table 5, we observe a clear convergence pattern: Merging just 3 to 5 prompts significantly improves generation quality (PSNR) compared to a single prompt, as the mask covers a broader range of potential attention patterns. In our final implementation, we use N=3 prompts, which offers an optimal balance between high sparsity (Efficiency) and high PSNR (Quality).

Table 5: Trade-off between Number of Merged Prompts, Sparsity, and Quality (PSNR). Data collected on Wan2.1 and Hunyuan.

| Prompts | HunyuanVideo | | Wan2.1-14B | |
|---|---|---|---|---|
| | Sparsity (%) ↑ | PSNR (dB) ↑ | Sparsity (%) ↑ | PSNR (dB) ↑ |
| 1 | 74.72 | 26.23 | 40.23 | 20.45 |
| 3 | 62.36 | 30.08 | 33.99 | 23.73 |
| 5 | 60.49 | 30.74 | 30.59 | 24.12 |
| 7 | 58.32 | 31.42 | 29.26 | 24.56 |
| 10 | 56.56 | 31.72 | 28.27 | 24.93 |

## H ROBUSTNESS OF STATIC MASKS

Whether masks searched on proxy prompts can generalize to unseen, highly dynamic scenes? We validate this through Attention Recall Analysis and Visualization.

### H.1 ATTENTION RECALL ANALYSIS ON DYNAMIC SCENES

We applied the static mask configuration (searched via 3 proxy prompts) to a set of unseen test videos, including highly dynamic scenes (e.g. "A racing car drifting", "A running taxi") and static scenes (e.g. "A chameleon"). We calculated the Recall Rate of the ground-truth attention values retained by our masks.

As illustrated in Figure 12a and Figure 12b, the heatmaps remain consistently high-valued across both static and dynamic scenarios. We also observed static scenes would mostly get more recall under the same static mask. However, the difference in recall between a Static and a Dynamic scene is less than 1%, proving that the union-mask strategy successfully captures the critical area of required attention regions.

### H.2 VISUALIZATION OF ATTENTION PATTERNS

To further elucidate the mechanism behind the robustness of static masks on dynamic scenes, we conduct a comparative visualization of raw attention maps in Figure 13.

**Visualization Methodology.** To capture the intrinsic behavior of attention heads, we visualize the raw attention scores from a specific query token against all key tokens in the video sequence. Specifically, we extract the Query ($Q$) and Key ($K$) tensors from a specific Layer during the generation process. We select a query token corresponding to a fixed spatial position in the middle frame (e.g., center of the canvas). We then compute the full attention map $\text{Attn} = \text{Softmax}(Q_{\text{query}} \cdot K^T / \sqrt{d})$. This resulting attention vector (shape $[1, T \times H \times W]$) represents how much the model "looks at"

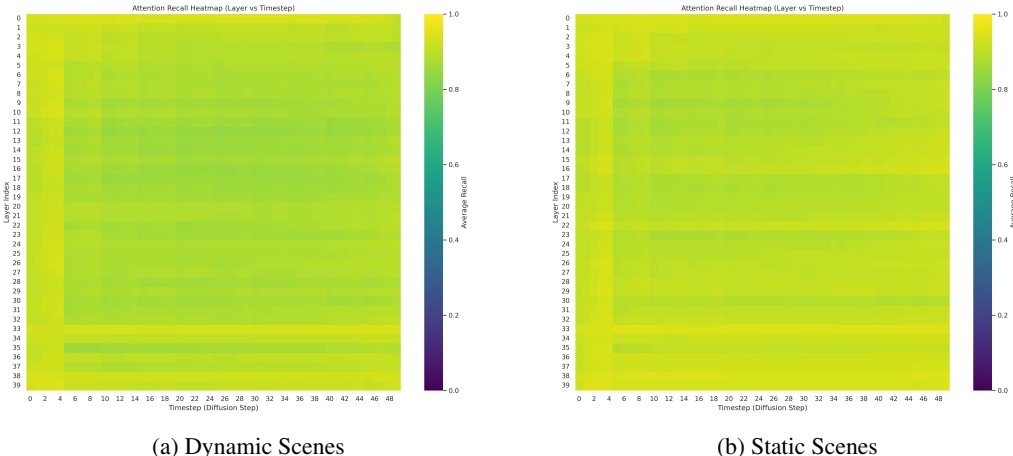

(a) Dynamic Scenes                                      (b) Static Scenes

Figure 12: **Attention Recall Heatmaps.** We visualize the recall rate of attention heads across different layers for (a) Dynamic Scenes and (b) Static Scenes. Brighter colors (yellow) indicate high recall (over 95%). The results show that our static masks consistently cover the activated attention regions even for intense motion, with an average recall exceeding 85% across all test cases.

every other pixel in the video when updating that specific query pixel. We reshape this vector back into video frames ($[T, H, W]$) and concatenate them vertically to form the heatmaps shown in Figure 13 (neighborhood frames were added as one so only 7 frames can be seen). Each row in the figure represents the attention distribution of a specific attention head.

**Functional Specialization over Semantic Dependency.** As observed in Figure 13, although the input prompts vary drastically—ranging from a static "drone camera" to a fast-moving "Taxi"—the spatial distribution of attention weights remains structurally consistent. For instance, specific heads in Layer 0 consistently exhibit "horizontal axial" patterns (attending to the same row) or "local block" patterns regardless of the object's position or motion speed. This indicates that many attention heads in Video DiTs evolve to perform fixed functional roles (e.g., aggregating spatial context along axes) rather than tracking specific semantic objects pixel-by-pixel.

**Justification for Compact Attention.** This observation directly validates our approach. Since the shape of the attention distribution (the "structural bias") is largely determined by the head's functional role rather than the dynamic input content, our Deformable Dual-Window design can effectively model these stable shapes. By capturing the union of these functional receptive fields offline, Compact Attention naturally accommodates dynamic motion without the need for expensive online adaptive searching.

# I  ADDITIONAL EXPERIMENT RESULTS

## I.1  COMPARATIVE ANALYSIS WITH BASELINE METHODS

Fig. 14 demonstrates the superior performance of Compact Attention compared with Sliding Tile Attention (STA) in the Hunyuan video generation framework. Our method achieves enhanced Peak Signal-to-Noise Ratio (PSNR) while operating at higher sparsity rates. Specifically, to accommodate STA's tile grouping requirements for attention sequence processing, we conducted comparative evaluations using 117-frame video sequences. The visual comparisons reveal that Compact Attention maintains better video quality preservation despite increased sparsity levels, confirming that our approach more effectively identifies and retains critical attention computation components - a core design principle of our architecture.More generation cases are shown in Fig. 15.

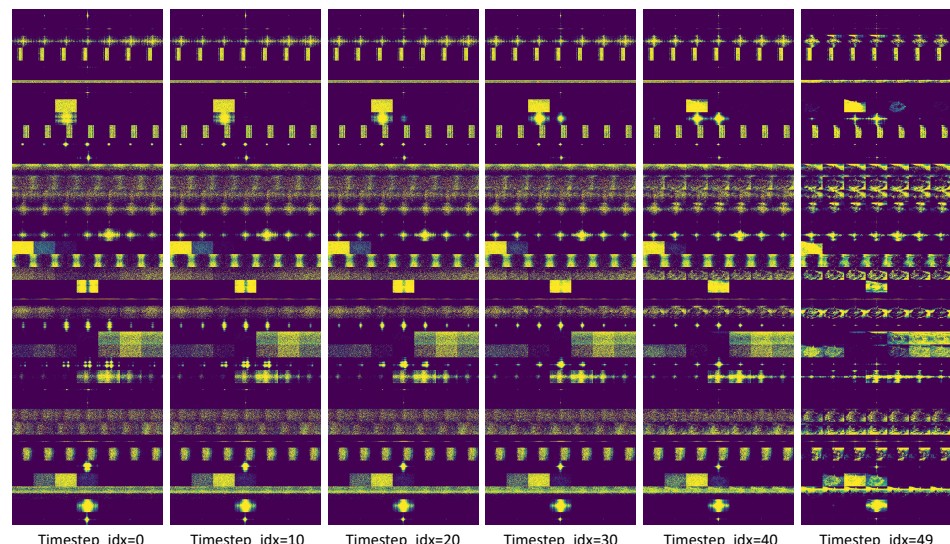

Timestep_idx=0   Timestep_idx=10   Timestep_idx=20   Timestep_idx=30   Timestep_idx=40   Timestep_idx=49

Wan2.1 14B: Static Scene Attention Map in Layer 0 with condition

Seed: 20218
Prompt: A drone camera circles around a beautiful historic church built on a rocky outcropping along
the Amalfi Coast, the view showcases historic and magnificent architectural details …

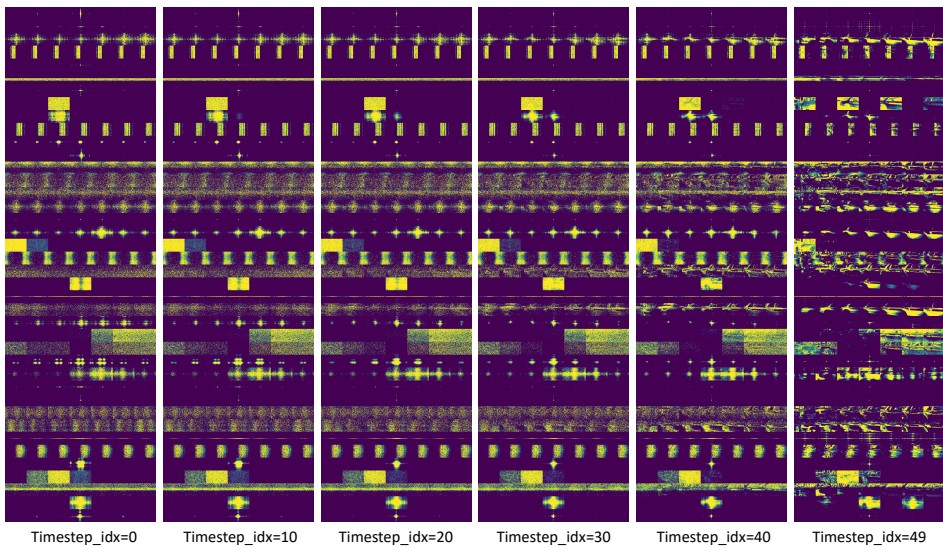

Timestep_idx=0   Timestep_idx=10   Timestep_idx=20   Timestep_idx=30   Timestep_idx=40   Timestep_idx=49

Wan2.1 14B: Dynamic Scene Attention Map in Layer 0 with condition

Seed: 578638
Prompt: A vibrant yellow taxi, its glossy surface reflecting city lights, speeds through bustling urban
streets, captured in a close-up shot that highlights its sleek design and polished exterior. …

Figure 13: **Visualization of Attention Pattern Persistence across Static and Dynamic Scenes.** We visualize the raw attention maps from Layer 0 of Wan2.1-14B at distinct denoising timesteps (idx=0 to 49). Each row corresponds to a specific attention head, showing its attention distribution across frames (concatenated horizontally). Brighter yellow areas indicate higher attention scores. **Top:** A static scene ("A drone camera"). **Bottom:** A highly dynamic scene ("A speeding Taxi"). Note that despite the drastic difference in motion, the active regions (yellow patterns) remain structurally similar for the same head index (e.g., horizontal lines or local blocks).

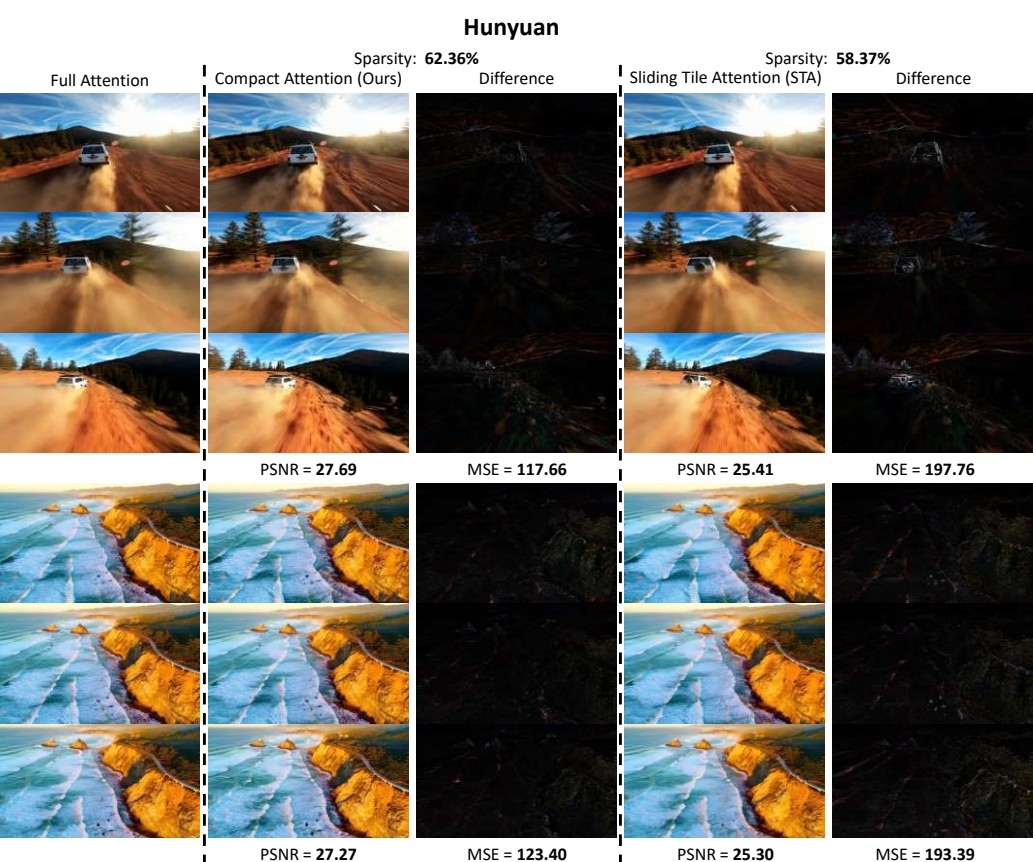

Figure 14: Performance of Compact Attention and Sliding Tile Attention on end-to-end video generation.

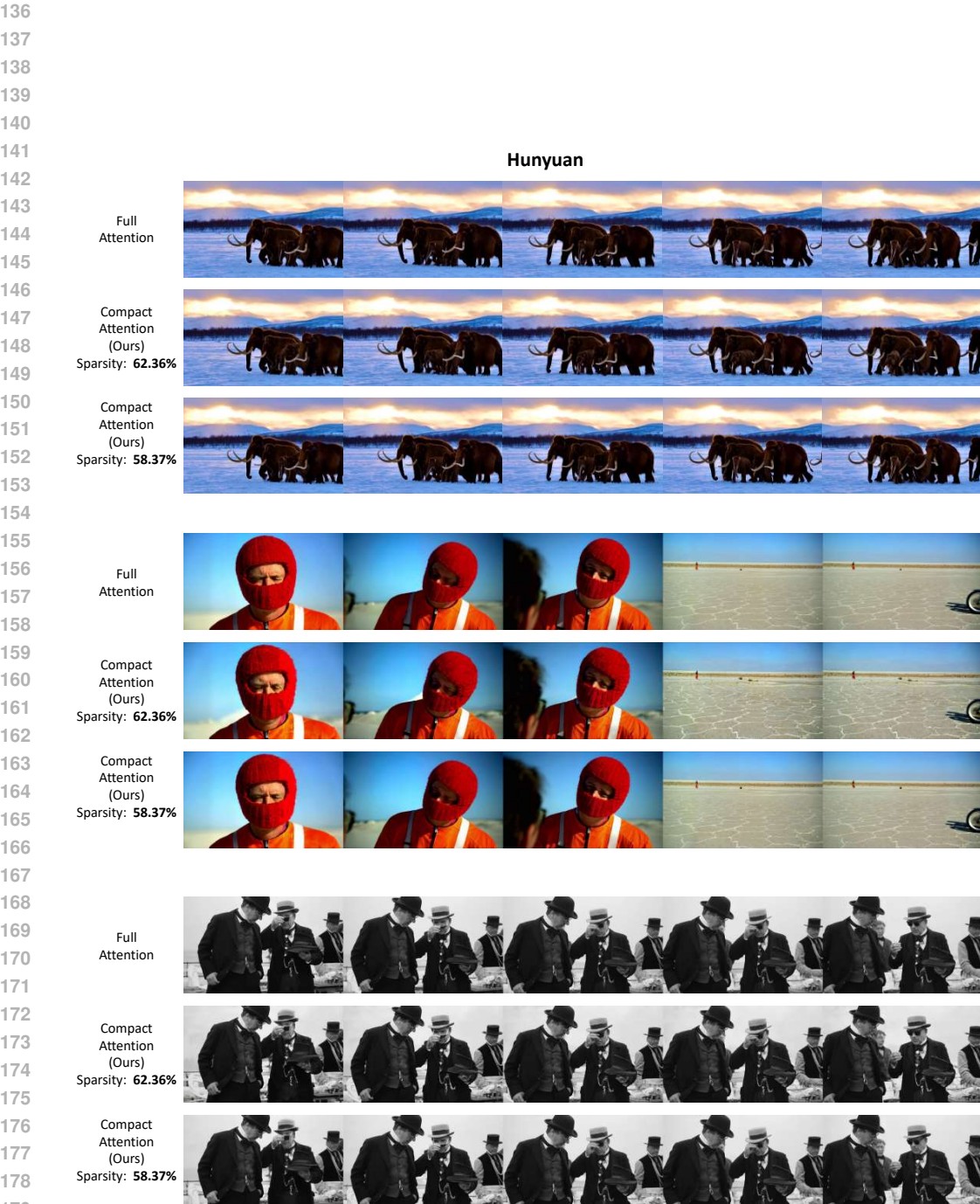

Figure 15: Performance of Compact Attention on end-to-end video generation.

