# OpenReview forum: "Compact Attention: Exploiting Structured Spatio-Temporal Sparsity for Fast Video Generation"
_ICLR.cc/2026/Conference — Submitted to ICLR 2026_

### Official Review · Reviewer_H3p5 · 2025-10-25

**Soundness:** 4
**Presentation:** 4
**Contribution:** 3
**Rating:** 8
**Confidence:** 4

**Summary:**

This paper proposes a training-free sparse attention framework that accelerates video diffusion transformers by exploiting stable spatiotemporal attention patterns. Through analysis of models like HunyuanVideo and Wan 2.1, the authors identify recurring cross-shaped spatial and time-dependent patterns and design a tile-based deformable mask with dual rectangular windows and frame-group-wise sparsity. An offline greedy search algorithm then optimizes these masks for efficiency while maintaining recall. Implemented on FlashAttention-3, Compact Attention achieves up to 3× faster attention computation and 2× end-to-end speedup with minimal quality loss, outperforming prior sparse attention methods such as STA, SVG, and SpargeAttention

**Strengths:**

1. The observation of cross-shaped attention patterns (horizontal and vertical spatial structures) is novel and complements prior findings that emphasized 3D cube (STA) or spatial-temporal periodic (SVG) patterns. This adds a valuable new perspective to understanding video transformer sparsity.
2. The iterative shrinking and auto-search mechanism is well-motivated and effectively designed. The greedy mask search algorithm demonstrates clear advantages over prior heuristic approaches such as STA’s manually pre-defined masks.
3. I appreciate that the authors report full-attention baselines using FlashAttention-3. Many previous works (and even other submissions I reviewed) play the trick of comparing against FA-2 on Hopper GPUs, which can exaggerate speedups. This transparency adds credibility.
4. The authors explicitly analyze the effect of keeping the first few denoising steps dense. This is an important implementation trick that several previous papers quietly adopt without quantifying its impact.

Overall, the paper presents solid results and comprehensive experiments with clear visual and quantitative evidence

**Weaknesses:**

1. The term “attention vocabularies” may be unnecessary. Maybe avoid constructing new terms?
2. The proposed method requires an offline greedy search per head and per frame-distance group, followed by a union across prompts. It would be nice if the paper quantify: a. he total GPU hours or wall-clock time required for this search b.how often the search must be repeated (e.g., for a new  resolution or different finetunes of the same base Wan 2.1 model).

**Questions:**

1. What would happen if one searched for a very high-sparsity mask and then finetuned the model to recover quality? In that case, might the first few steps no longer need to remain dense? (This is not a required experiment but would be interesting to discuss.)
2. How exactly is the sparsity in Tables 1 and 2 computed? Does it account for the early full-attention steps, or is it measured only over the sparse steps?

---

> ### Author Response · Authors · 2025-11-25
>
> We are grateful for your strong support and excellent summary of our contributions.
>
> W1: "Attention Vocabularies".
>
> **A:** We agree "Attention Vocabularies" could be not appropriate. We have changed this in the revision.
>
> W2: Search Cost and Finetuning.
>
> **A:** Thanks for your constructive advice, we have added a Section to discuss about the cost of offline search. As detailed in General Response 3, the search time is **~1.5 hours** on a 8 H800s for the model using one proxy prompt for Wan2.1-14B. This is a one-time pre-computation. The search needs to be repeated if the **model architecture** or **aspect ratio** changes or used on different finetunes (e.g., LoRAs) of the *same* base model.
>
> **For Finetuning:** This is an exciting direction.Currently, we retain dense attention in early steps because the latent space is dominated by noise, making sparse patterns less effective. **Sparse Finetuning** could potentially adapt the model to function with sparse attention even in these early noisy stages, eliminating the need for dense steps. We leave this for future work.
>
> Q2: Sparsity Calculation.
>
> **A:** The reported sparsity in tables are the **global average** over the entire generation process, including the initial full-attention steps. This ensures the quality of videos correspond directly to the reported sparsity.

---

### Official Review · Reviewer_DMSP · 2025-10-29

**Soundness:** 2
**Presentation:** 2
**Contribution:** 2
**Rating:** 2
**Confidence:** 5

**Summary:**

The paper proposes Compact Attention, a training-free video generation acceleration method based on the spatial and temporal redundancy. It mainly consists of three components:

1. Adaptive sparse tiles, which allows different sparsity levels for near and distant frames with the proposed spatial tiles.

2. Offline search for mask: the authors analyze the attention maps during the video generation for different prompts, and provide a static sparse strategy for the future inference usage.

This work achieves better performance than several baselines and achieves up to 3x acceleration.

**Strengths:**

1. This work shows that the proposed method achieves better performance than several baselines.

2. This work shows that they can achieve up to 3x acceleration.

**Weaknesses:**

1. The novelty of spatial and temporal redundancy focus is limited, which is similar as SVG [1] . SVG also partitions tokens into local tiles, and employs cross-frame attention masks.

2. The offline search for the optimal static pruning strategy is empirically unconvincing. The sparse patterns always related to the input prompt, denoising step, layer depth, and seed. The author also show that the similarity is over 0.8, which means the static strategy may not be the optimal one and may lead to crash in some special case.

3. In Table 1, the experiments are not clear, the sparsity is wired. The sparsity used for different methods are different, and the similarity metrics including SSIM, PSNR, and LPIPS are not reported.

4. In Table 2, again, the sparsity is not equal, it is a unfair comparison, and the SVG results are not included.

----
[1] Sparse VideoGen: Accelerating Video Diffusion Transformers with Spatial-Temporal Sparsity

**Questions:**

1. Please provide the clear experimental results compared to other baselines, the results should include the similarity metrics.

2. Why the proposed method claims for the dynamic sparsity? the off-line search is adopted to get a static pruning strategy.

---

> ### Author Response · Authors · 2025-11-25
>
> We thank the reviewer for the critical feedback.
>
> W1: Limited Novelty compared to Sparse VideoGen (SVG).
>
> **A:** We respectfully disagree. While both works utilize sparsity, **Compact Attention differs fundamentally in "Pattern Granularity" and "Search Mechanism",**  offering a superior trade-off through **Granularity** and **Precision.**
>
> 1. **Pattern Granularity: Deformable dual-window vs. Binary:** SVG relies on a coarse binary classification (Local Block vs. Temporal Stride), and different attention heads share two fixed, predefined masks with a same sparsity. Our dual**-**window design flexibly approximates different patterns and apply masks with different sparsity, which allow us to accelerate those attention heads who only need masks with high sparsity while protecting the others.
> 2. **Search Mechanism: Greedy offline search vs. Online decision:** SVG would sample and determine a better mask online during generation, while Compact Attention utilizes an offline search strategy and apply static masks. Offline search largely reduces overhead of mask decision, enables greater speedup.
>
> **W2: Offline search is empirically unconvincing**
>
> **A:** The similarity between prompts reflects raw intersection between single-prompt masks. However, our deployment uses the **Union** of these masks, which is designed to handle difference of single-prompt masks in special cases. We added a Section in Appendix to discuss the robustness of static mask in various scenes. There we also show attention maps in different circumstances which you might find interesting.
>
> W3, W4 and Q1：**Missing Similarity Metrics and Fair Comparison.**
>
> **A:** Different sparse methods can hard to be strictly same for different strategies are used. For example, SpargeAttn adopts dynamic masks and we reported the average sparsity rate. To enable fair comparison, we tuned Compact Attention thresholds to match the sparsity levels of baselines and stay higher to compare.  For missing similarity metrics, we added results to Table 2 in the paper.
>
> Q2: Why the proposed method claims for the dynamic sparsity?
>
> **A:** Our method did not claims for dynamic sparisty. “Dynamic sparsity” mentioned refers to the sparsity of optimal mask for each attention head to remain important information during a generation. Masks searched offline are static for a single attention head but are various between attention heads or timesteps.

---

> > ### Comment · Reviewer_DMSP · 2025-11-25
> >
> > Thanks for the rebuttal.
> >
> > How is the sparsity computed in the Table 2? As according to work [1], the work SVG [2] achieves good similarity results (PSNR > 25) when the sparsity is 55%. However, in Table 2 of this work, for 25% sparsity, the PSNR is 18 with Wan2.1 model
> >
> > [1] DraftAttention: Fast Video Diffusion via Low-Resolution Attention Guidance
> >
> > [2] Sparse VideoGen: Accelerating Video Diffusion Transformers with Spatial-Temporal Sparsity

---

> > > ### Author Response · Authors · 2025-11-25
> > >
> > > We thank the reviewer for the close examination of our experimental results. We clarify the sparsity calculation and the performance observations for the Sparse VideoGen (SVG) baseline on the Wan2.1 model below.
> > >
> > > **1. How Sparsity is Computed in Table 2**
> > >
> > > In Table 2, the "Sparsity" column represents the **Global Average Sparsity** (the percentage of pruned attention calculation) over the entire generation process compared to the Full Attention baseline (0%).
> > >
> > > To ensure a fair evaluation using the official SVG codebase, we configured the script with setting arg “height” as 768 and arg “sparsity” as 0.65 (which we think is used as “density” ). Other args are not modified. And then SVG will generate block mask for flex attention and the actual measured sparsity was **32.08%**.
> > >
> > > Our reported figure of **25.02%** is derived as follows:
> > >
> > > - **Step Schedule:** Total 50 steps. The first **10 steps** are Full Attention (Dense). The remaining 40 steps use Sparse Attention.
> > > - **Layer Schedule:** Total 40 layers. The **first layer** is kept as Full Attention (Dense). The remaining 39 layers use Sparse Attention.
> > > - **Block Sparsity:** Measured at ~32.08% (pruned ratio).
> > >
> > > The calculation for the Global Sparsity is:
> > > Global Sparsity = 32.08\% * (39/40) * (40/50) = 25.0224\%
> > >
> > > **2. Regarding Performance mismatch**
> > >
> > > We have carefully reviewed [1] and attribute the performance difference to two factual reasons: Metric Definition and Implementation Variance.
> > >
> > > Paper [1] states they *"retain full attention... for the first 25% of denoising steps"*. Their reported sparsity ratios (e.g., 55%) likely refer to the **sparsity during the sparse stage**, excluding the dense stage. In contrast, our reported **25.02%** is the **Global Average Sparsity** (factoring in the 10 dense steps and the first dense layer). And we noticed keeping the first layer dense was not reported in [1]. So implement of SVG in [1] is not quite clear.
> > >
> > > > [1] DraftAttention: Fast Video Diffusion via Low-Resolution Attention Guidance
> > > >
> > >
> > > We hope this clarifies the calculation and the context of the baseline performance.

---

### Official Review · Reviewer_6MNW · 2025-10-30

**Soundness:** 2
**Presentation:** 2
**Contribution:** 3
**Rating:** 4
**Confidence:** 4

**Summary:**

This paper proposes Compact Attention. The authors observe that for the same DiT model, the attention patterns remain similar across different inputs. Based on this, they precompute the attention patterns offline and apply them during video generation. Specifically, the method represents sparse masks using Dual Attention Windows and employs a greedy algorithm for the offline precomputation. Experiments show that Compact Attention can achieve up to 3× attention speedup in video generation while maintaining video quality.

**Strengths:**

The proposed approach is novel, particularly the idea of offline precomputation of sparse masks and the use of dual attention windows to represent attention masks.

The kernel implementation is tailored for their method.

**Weaknesses:**

The paper’s presentation is rather unclear in several places. For example:

- The description of the greedy algorithm suggests a progressive contraction, but the pseudocode shows progressive expansion.
- In Figure 4, there is a “Flag” term that is never defined in the text.
- Sections such as Reuse Masks across Denoising Steps lack sufficient details.


The experimental evaluation is limited. For instance, in Table 2, on Wan 2.1, the STA method achieves twice the sparsity of Compact Attention, which makes the performance comparison less convincing.

The paper statistically verifies that the sparse patterns in DiT models are independent of the prompt and random seed. This finding is somewhat counterintuitive and lacks sufficient analysis and discussion.

**Questions:**

The authors focus on three spatial patterns and fit them using the dual attention windows. However, this design choice lacks justification. Could the authors provide quantitative evidence showing what proportion of attention maps actually exhibit these three spatial patterns?

What is the computational cost of the offline greedy algorithm, and how much memory is required to store the precomputed attention patterns?

In the Quality Evaluation section, the authors argue that video similarity should not be compared, yet in the comparison with STA, only similarity metrics are reported. Why not use the quality evaluation metrics introduced in that section to compare against STA?

Sparse video gen has a 2nd version; I suggest that you use it as a baseline.

---

> ### Author Response · Authors · 2025-11-25
>
> We appreciate your detailed review and recognition of our novel offline strategy and kernel implementation.
>
> W1: Presentation Clarity.
>
> **A:**  We apologize for the confusion and have revised the text:
>
> 1. Algorithm:  The pseudocode in the Appendix has been corrected to match the text.
> 2. "Flag": In Fig. 4, Flag is a boolean indicator. Flag=1 indicates a mask candidate is still active for shrinking; Flag=0 means it has reached the quality/cost limit. We added this definition to the caption.
> 3. Mask Reuse: We clarify that masks are reused for **5 consecutive steps**, as patterns drift slowly.
>
> W2: Less convincing performance comparison.
>
> A: We acknowledge the concern. The official STA configuration yields suboptimal quality when forced to higher sparsity levels due to its rigid block structure. As detailed in General Response 2,  Ablation Sec 5.3 shows Compact Attention performs better than methodology like single-window strategy and mask searching from predefined mask candidates.
>
> W3: Independence of Prompt/Seed is counterintuitive.
>
> **A:** We agree it is surprising. While specific activation intensities vary with prompts and seeds, the *required* spatial coverage for fine video generation is predictable. And some attention heads even show stable obvious spatial patterns independent of prompt and seed. This may stem from the **architectural bias** of Video DiTs and effect by RoPE. To make further analysis and discussion,  we have expanded **Appendix H** to discuss this "Functional Specialization" hypothesis.
>
> Q1: Proportion of patterns.
>
> **A:** Three spatial patterns are not strictly defined for each attention head but combined to reach higher effectiveness in the final mask configurations.  Here we show the proportion of heads based on their final mask shapes in the first temporal group.
>
> |  | Local Patterns | Cross-Shape Patterns | Global Patterns | Time-Variant Patterns | Time-Invariant Patterns |
> | --- | --- | --- | --- | --- | --- |
> | Wan2.1 | 14.49% | 67.00% | 18.52% | 20.57% | 79.43% |
> | Hunyuan | 15.35% | 65.41% | 19.24% | 34.63% | 65.37% |
>
> Q2 and Q3 are answered in global response.
>
> Q4: Suggestion of SVG2 as a baseline
>
> **A:** We added SVG2 as a baseline in the paper, please refer to Table 1&2.

---

### Official Review · Reviewer_gfd6 · 2025-11-02

**Soundness:** 3
**Presentation:** 3
**Contribution:** 2
**Rating:** 4
**Confidence:** 3

**Summary:**

The paper introduces Compact Attention, a framework designed to accelerate transformer-based video generation by exploiting structured spatio-temporal sparsity in attention mechanisms. The authors identify recurring sparse attention patterns in video diffusion transformers—such as cross-shaped and time-variant structure and propose a sparse attention mechanism based on adaptive tile-based sparse masks and a greedy auto-search algorithm that precomputes optimal attention masks. Compact Attention achieves up to 3× acceleration on large-scale models like Hunyuan and Wan2.1 while maintaining comparable visual quality and temporal consistency.

**Strengths:**

1. The authors demonstrates significant acceleration (up to 3×) with negligible quality loss, validated on strong baselines and realistic benchmarks.

2. The method is training-free and hardware-aware, making it practical for real-world applications and compatible with existing transformer architectures.

3. The study on the effectiveness on "delaying sparse attention" is good. Although it is well-observed in previous studies, none of them plot this trend.

**Weaknesses:**

1. The offline mask search, though effective, could be computationally heavy for deployment across diverse configurations. The authors should add experiments to discuss how the final quality changes with respect to the computation spent based on this calibration process.

2. On highly dynamic or non-redundant video scenes, is the sparsity pattern still highly similar to the offline searched sparse attention pattern? It seems that in Figure 3(b) some similarity score is as low as 0.7 (based on color).

3. The authors should faithfully report the speedup number for Wan 2.1 and HunyuanVideo separately, rather than only reporting "up to 3x".

**Questions:**

Please refer to the weakness section.

---

> ### Author Response · Authors · 2025-11-25
>
> Thank you for acknowledging the practical value of our training-free, hardware-aware framework and our analysis of delaying sparse attention.
>
> W1: Offline mask search overhead and deployment.
>
> **A:** As detailed in General Response 3, the search time is **~1.5 hours** on a 8 H800s for the model using one proxy prompt for Wan2.1-14B. This is a one-time pre-computation. Since this is a one-time offline process and it enables parallel searching, this setup cost is amortized to near-zero. We have added a "Cost and Convergence Analysis of Offline Search" section in the Appendix G.
>
> W2: Dynamic scenes and pattern similarity.
>
> **A:** The low similarity (0.7) in Fig 3(b) reflects raw intersection between single-prompt masks. However, our deployment uses the **Union** of these masks.
>
> We added “Recall Analysis” Section in Appendix comparing attention heatmaps and attention recall under Compact Attention of a "Moving Taxi" (dynamic) vs. "Chameleon" (static). It shows that the "Union Mask" effectively covers the critical active regions of the dynamic scene, as the attention heads activate structurally similar areas despite object motion.
>
> W3: Separate speedup reporting.
>
> **A:** We have updated the abstract to be precise:
>
> **Wan2.1 (14B):** 2.25x attention speedup. **HunyuanVideo:** 3.03x attention speedup.

---

### Author Response · Authors · 2025-11-25
**General Response to All Reviewers: Robustness, Efficiency, and Comparisons**

We thank all reviewers for their insightful feedback and for recognizing our method's effectiveness (achieving significant speedup) and practical value (training-free, hardware-aware). We have updated the manuscript to incorporate the suggested baselines and clarifications. Below, we address the three primary concerns shared across reviews.

1. Why Static Masks Work on Dynamic Scenes (Robustness)

***Response:*** Reviewers (gfd6, DMSP) questioned if offline-searched static masks can handle highly dynamic videos.

The vbench score of end-to-end video generation is a strong evidence of Compact Attention's generalization across different scenes. To show the result is not merely an average score eliminating failure cases. We also report results of more analysis on attention patterns.

Our extensive analysis reveals that attention heads in DiTs exhibit **"Functional Specialization"** (e.g., attending to horizontal axes, local neighborhoods, or specific temporal distances) rather than tracking semantic objects pixel-by-pixel. This can be staightly observed through the new **Figure 13**, which we are eager to show to all readers. Attention heads show high attention score in similar regions cross different prompts.

We also conducted a **Recall Analysis** on unseen dynamic scenes (e.g., "Racing Car", "Moving Taxi") and static scenes (e.g., "Chameleon", "Coast"). As shown in the new **Appendix H**, our "Union Strategy" (merging masks from proxy prompts) covers **>88%** of ground-truth attention mass in dynamic scenes, virtually identical to static scenes (>90%).

***Conclusion:*** Although counterintuitive, the structural patterns of attention are intrinsic to the model/layer/timesteps, making static masks robust across diverse motion dynamics. This is where we see potential of accelerating attention calculation through static masks.

2. Fairness of Comparison vs. Baselines

***Response:*** Reviewers (6MNW, DMSP) asked about comparisons with STA and other baselines.

***Iso-Sparsity Comparison:*** In the revised Table 1 & 2, we compared Compact Attention against baselines under strict iso-sparsity or higher sparsity settings. Our method consistently achieves higher VBench scores and similarity metrics.

***Note regarding STA:*** We utilized the official STA configurations for comparison as details of their search code is unavailable. We also clarify that our Frame-group-wise Dual-Window design allows for higher sparsity than STA’s single-window approach, Greedy search algorithm also has better performance compared with choice from predefined mask candidates (see Ablation Sec 5.3).

3. Offline Search Overhead is Negligible

***Response:*** Reviewers (gfd6, H3p5) inquired about the cost of the auto-search algorithm.

The search is a **one-time** offline cost per model resolution (approx. 1.5 hours on 8xH800 for Wan2.1-14B). The resulting mask config is tiny (<3MB). In deployment scenarios serving millions of requests, this one-time initialization cost is amortized to near-zero. Please refer to **Appendix G** for a detailed "Cost and Convergence Analysis of Offline Search"

---

### Meta-Review · Area_Chair_AyXK · 2025-12-24

**Summary:**

The paper proposes a sparse attention method for video generation. The method observes a sparsity pattern different than existing methods such as SVG, and only requires an offline search of sparsity masks, which can be more efficient than existing methods which requires online profiling. Speed results is reported on H800 with highly optimized dedicated kernels for Hopper. Up to 3x speedup is reported, though a large proportion of speedup may come from Hopper-specific kernel optimizations.

Reviewers have concerns on the methodology, about whether using a fixed sparsity pattern for all prompts and seeds makes sense. Furthermore, there are concerns about the faithfulness of the experimental results. Particularly, the performance of baselines is significantly lower than original papers / other literature. Due to the remained concerns, I recommend a rejection.

**Reviewer Concerns:**

1. whether the static mask can transfer to highly dynamic videos (gfd6) / why the sparsity pattern is independent of prompts (6MNW)
2. experimental issues, such as isosparsity comparison and psnr results.
3. the results of SVG and other baselines deviates significantly from literature (DMSP)

While the first to concerns are supported by arguments, new results and revisions. AC think the third concern can be significant. It is quite weird why both SVG and SVG2 have such a low PSNR <20 even at a very low sparsity level ~30%. This may indicate potential mismatches in reproducing existing works and may invalidate the superior sparsity-quality tradeoff of the paper.

**Reviewer Scores:**

Original rating is (4,4,2,8). I think reviewers will retain their original rating.

---

### Decision · Program_Chairs · 2026-01-26

Reject